# Identification and validation of a m7G-related lncRNA signature for predicting the prognosis and therapy response in hepatocellular carcinoma

**Yue-Ling Peng[1], Ya-Fang Dong[2], Li-Li Guo[3], Mu-Ye Li[4], Hui Liao[5], Rong-Shan Li[1]***

1 Department of Nephrology, Shanxi Provincial People's Hospital (Fifth Hospital of Shanxi Medical University), Taiyuan, China, 2 Department of Pathology and Pathophysiology, School of Basic Medicine, Shanxi Medical University, Taiyuan, China, 3 Provincial Key Laboratory of Nephrology, Shanxi Provincial People's Hospital (Fifth Hospital of Shanxi Medical University), Taiyuan, China, 4 Department of Ocular Fundus Diseases, Shanxi Eye Hospital, Shanxi Medical University, Taiyuan, China, 5 Drug Clinical Trial Institution, Shanxi Provincial People's Hospital (Fifth Hospital of Shanxi Medical University), Taiyuan, China

* rongshanli@126.com

**Data Availability Statement:** Publicly available datasets from The Cancer Genome Atlas (https://portal.gdc.cancer.gov/) were analyzed in this study.

## Abstract

### Background

N7-methylguanosine (m7G) is one of the most common RNA posttranscriptional modifications; however, its potential role in hepatocellular carcinoma (HCC) remains unknown. We developed a prediction signature based on m7G-related long noncoding RNAs (lncRNAs) to predict HCC prognosis and provide a reference for immunotherapy and chemotherapy.

### Methods

RNA-seq data from The Cancer Genome Atlas (TCGA) database and relevant clinical data were used. Univariate and multivariate Cox regression analyses were conducted to identify m7G-related lncRNAs with prognostic value to build a predictive signature. We evaluated the prognostic value and clinical relevance of this signature and explored the correlation between the predictive signature and the chemotherapy treatment response of HCC. Moreover, an in vitro study to validate the function of CASC19 was performed.

### Results

Six m7G-related lncRNAs were identified to create a signature. This signature was considered an independent risk factor for the prognosis of patients with HCC. TIDE analyses showed that the high-risk group might be more sensitive to immunotherapy. ssGSEA indicated that the predictive signature was strongly related to the immune activities of HCC. HCC in high-risk patients was more sensitive to the common chemotherapy drugs bleomycin, doxorubicin, gemcitabine, and lenalidomide. In vitro knockdown of CASC19 inhibited the proliferation, migration and invasion of HCC cells.

The R code used can be found at https://doi.org/10.6084/m9.figshare.21830130.v1.

**Funding:** This study was supported by the Natural Science Foundation of Shanxi Province (201903D421020) and the Special Fund of Central Government Guided Local Scientific Development (YDZJSX2021O027). There was no additional external funding received for this study. The funders had no role in study design, data collection and analysis, decision to publish, or preparation of the manuscript.

**Competing interests:** The authors have declared that no competing interests exist.

**Abbreviations:** CASC19, cancer susceptibility 19; GSEA, gene set enrichment analysis; m7G, N7-methylguanosine; PCA, principal component analysis; ROC, receiver operating characteristic; TIDE, tumor immune dysfunction and exclusion.

## Conclusion

We established a 6 m7G-related lncRNA signature that may assist in predicting the prognosis and response to chemotherapy and immunotherapy of HCC.

## Introduction

Liver cancer is the sixth most prevalent cancer and the fourth leading cause of cancer-related death globally [1]. Hepatocellular carcinoma (HCC) currently accounts for approximately 90% of all primary liver malignancies and is a serious worldwide health issue with increasing incidence [2]. Despite considerable progress in treatment strategies, the survival rate of patients with HCC is far from satisfactory due to a low rate of early detection, a tendency for recurrence, and chemotherapy resistance [3]. Therefore, comprehensive and in-depth studies are required to identify new therapeutic strategies.

According to the 2021 MODOMICS study, more than 300 distinct posttranscriptional chemical alterations have been detected in RNAs [4]. Methylation is involved in almost every step of RNA metabolism [5, 6]. RNA methylation has been reported to be associated with a wide range of physiological and disease conditions, while abnormal methylation can lead to disease and cancer [7–9]. N7-methylguanosine (m7G) is the most prevalent modification in the 5′ cap of mRNA [10] but has also been identified in several internal mRNA, tRNA, and rRNA sites [11–13]. A growing number of studies have indicated that m7G modification has extensive effects on mRNA, tRNA, and rRNA and plays a key role in oncogenic mRNA translation and cancer development. According to Ying X, the METTL1-m7G-EGFR/EFEMP1 axis can enhance bladder cancer growth [14]. Xia P also reported that MYC-targeted WDR4 enhances growth, metastasis, and sorafenib tolerance by increasing CCNB1 translation in HCC cells [15]. Recent research in lung cancer has revealed new evidence regarding mRNA translation modulation via tRNA changes and the related mRNA codon compositions [16]. In addition, a relationship among METTL1-mediated m7G tRNA alteration, oncogenic mRNA translation regulation, and cancer development has also been identified in intrahepatic cholangiocarcinoma [17].

Long noncoding RNAs (lncRNAs) are transcripts that contain more than 200 nucleotides but are not translated into proteins [18]. lncRNAs have been proposed to carry out various cellular and physiologic functions and have been identified to be closely involved in carcinogenesis and tumor development [19–21].

Additionally, RNA methylation of lncRNAs has been proven to impact tumor development. For example, the lncRNA THAP7-AS1, transcriptionally initiated by SP1 and posttranscriptionally regulated by METTL3-mediated m6A alteration, improves CUL4B entry into the nucleus in gastric cancer [22]. Recent studies have revealed that the m5C-modified H19 lncRNA increases the relapse and development of tumors in HCC [23]. However, few investigations into the regulation of m7G in lncRNAs have been conducted to date, and the particular expression patterns and clinical value of m7G-related lncRNAs in HCC remain unclear.

In this study, for the first time, we established a predictive signature based on m7G-related lncRNAs as prognostic biomarkers. To develop new therapeutic methods for the treatment of HCC, we investigated the diagnostic and prognostic value and impact on immune checkpoint expression, chemotherapy sensitivity, tumor immune infiltration, and functions of RNA methylation of m7G-related lncRNAs in HCC.

## Materials and methods

### Data acquisition

A total of 47 m7G-related genes were identified in GeneCards (https://www.genecards.org/), and these genes are listed in S1 Table. The RNA-seq expression and clinical information for HCC patients were downloaded and extracted from The Cancer Genome Atlas (TCGA, https://cancergenome.nih.gov/). A total of 374 HCC samples and 50 normal liver tissues were acquired, with 343 cases containing complete clinical information and follow-up time.

### Identification of differentially expressed m7G-related genes and functional enrichment analysis

The "limma" package in R (4.0.3) was used to obtain the differentially expressed m7G-related genes (|logFC|>1, FDR < 0.05) in the HCC and normal groups. To further analyze the possible function and signaling pathways of m7G-related genes in HCC patients, Gene Ontology (GO) and Kyoto Encyclopedia of Genes and Genomes (KEGG) [24–26] pathway enrichment analyses were conducted using the "clusterProfiler" and "org.Hs.eg.db" R packages with P values of 0.05 and q-values of 0.05 as thresholds (Copyright permission was granted by Kanehisa Laboratories).

### M7G-related lncRNAs

A systematic bioinformatics approach was employed to identify m7G-related long non-coding RNAs (lncRNAs). Expression profiles for m7G-modified genes and lncRNAs were obtained from our dataset. We then computed the Pearson correlation coefficient between expressions of m7G-modified genes and lncRNAs. An absolute Pearson correlation coefficient ($|R^2|$) greater than 0.3 and a P-value less than 0.001 were deemed significant [27–30]. This strict criterion ensured the identification of lncRNAs exhibiting the strongest associations with m7G-modified genes. The method captured both positive and negative correlations. Finally, significantly correlated m7G-related lncRNAs were further investigated for potential biological functions and clinical relevance. Cytoscape 3.8.2 software was next used to visualize coexpression networks.

### Construction and validation of the m7G-related lncRNA predictive signature

We used univariate Cox analysis (P<0.05) based on m7G-related lncRNAs to assess the prognostic significance of candidate lncRNAs in HCC. Next, a multivariate Cox analysis (P<0.05) was used to establish a predictive signature for m7G-related lncRNAs. Each patient's risk score was computed using the following formula:

$$Risk\ score = (coef\ (lncRNAi) \times expr\ (lncRNAi)$$

HCC patients were separated into high- and low-risk groups based on their median risk score. Kaplan–Meier analysis was performed to compare the survival time of the subgroups. Additionally, we constructed receiver operating characteristic (ROC) curves for 1-year, 3-year and 5-year survival to examine the prognostic signature's prediction accuracy.

To further verify the stability of this prediction model, the whole dataset was used for internal validation. The TCGA-LIHC dataset was randomly divided into two cohorts: internal cohort 1 (n = 172) and internal cohort 2 (n = 171).

## Development and validation of the nomogram

Initially, a multivariate Cox regression analysis was performed, incorporating the risk score and clinicopathological parameters, to identify independent prognostic factors. These factors were then used to construct the prognostic nomogram using the "rms" package in R. For the development of the nomogram, each individual prognostic factor identified through the Cox regression analysis is assigned a weighted score in the nomogram, proportional to the magnitude of its regression coefficient. The sum of these scores forms the Total Points, which can be converted to a probability of survival at specified time points. Furthermore, calibration plots were generated to visually assess the predictive accuracy of the nomogram. In these plots, the x-axis represents the predicted survival probability by the nomogram, and the y-axis represents the actual observed survival. A 45-degree line in the plot represents perfect calibration, where predicted probabilities match the actual outcomes. The closer the calibration curve is to this line, the better the predictive accuracy of the nomogram.

## Gene set enrichment analysis

To investigate the biological pathways of the subgroups, we further performed gene set enrichment analysis (GSEA) for functional enrichment analysis. We employed the GSEA software from the Broad Institute, which allows the use of publicly available gene set databases, such as the Molecular Signatures Database (MSigDB). When performing GSEA, we first ranked our gene expression data, with this ranking based on the differential expression of genes across the different subgroups. Subsequently, we calculated an enrichment score (ES) for each gene set, reflecting the distribution of that set's genes within the ranked list. To determine the significance of the enrichment for each gene set, we generated a null distribution by random permutations of the original data and calculated a normalized enrichment score (NES) and normalized P-value for each set against this distribution. We then corrected these P-values to q-values using the False Discovery Rate (FDR) method to control for the probability of false discoveries when conducting multiple hypothesis tests.

## Immune activity analysis

To evaluate the link between the m7G-related lncRNA predictive signature and immune infiltration in HCC, single-sample gene set enrichment analysis (ssGSEA) was used to determine the infiltration scores of 16 immune cells and the function of 13 immune-related pathways utilizing "GSVA" software. After obtaining the expression of immune infiltration in each sample, Spearman's test was used to explore the correlation of the risk score with immune infiltration and immune-related functions. The immune cells were as follows: activated dendritic cells (aDCs); B cells; CD8 T cells; DCs; immature DCs (iDCs); macrophages; mast cells; neutrophils; natural killer cells (NK cells); plasmacytoid DCs (pDCs); T helper cells; T follicular helper (Tfh) cells; Th1 cells; Th2 cells; tumor infiltrating lymphocytes (TILs); regulatory T cells (Tregs). We also investigated the relationship between the risk score and immune checkpoint expression in HCC patients, considering the importance of immune checkpoint-related gene expression levels in immune checkpoint inhibitor therapy.

## Chemotherapy drug response prediction based on the m7G-related lncRNA signature

To assess the significance of the predictive signature in predicting the sensitivity of HCC to chemotherapy, the "pRRophetic" package was used to calculate the half-maximal inhibitory concentration (IC50) of the main chemotherapeutic medications used to treat HCC.

## Prediction of response to immunotherapy

The Tumor immune dysfunction and exclusion TIDE method is designed to model two primary mechanisms of tumor immune evasion that can affect the response to immunotherapy: the dysfunction of tumor-infiltrating cytotoxic T lymphocytes (CTLs) and the prevention of CTL infiltration. This model uses pre-existing transcriptomic data to generate predictions about the likelihood of response to immune checkpoint blockade treatment [31]. In our study, we obtained transcriptomic data of HCC patients from TCGA and stratified these patients into different risk groups based on their expression profiles. We then applied the TIDE method to each of these risk groups. The TIDE method uses the gene expression profile to calculate a TIDE score, which reflects the likelihood that the cancer will respond to immunotherapy. The TIDE prediction model is based on the principle that if the expression of a particular set of genes (known as a gene signature) associated with T cell dysfunction and exclusion is high, the tumor is more likely to evade the immune system, and hence the patient is less likely to respond to immune checkpoint blockade therapy.

## Cell culture and transfection

The human hepatoma cell line Hep3B was purchased from the Cell Bank of the Chinese Academy of Sciences, and Hep3B cells were grown in 90% DMEM containing 10% FBS (Gibco, USA) at 37°C, 95% humidity, and 5% $CO_2$ in a cell incubator. Two siRNA sequences targeting CASC19 and the nonspecific siRNA sequence (si-NC) were designed and produced by Gene-Pharma (Shanghai, China). The siRNA sequences of CASC19 were as follows: siCASC19-2, 5′-GCUCAGCAUUUGCCAUACUTT-3′; siCASC19-139, 5′-GCUUCCUAAAGAGAUAACATT-3′; and si-NC, 5′-UUCUCCGAACGUGUCACGUTT-3′. Following the manufacturer's instructions, transfections were conducted using YosiTrans R-100 RNA Transfection Reagent (Yoshi, China). The medium was replaced with fresh medium 6 hours after transfection, and cells were harvested 48 hours for further experiments.

## RNA extraction and real-time quantitative PCR (RT–qPCR) analysis

RNA was isolated using TRIzol reagent (Takara, Japan) and reverse transcribed to cDNA using PrimeScript RT Master Mix (Takara, Japan). The RT–qPCR analyses were performed in triplicate using the NovoStart SYBR qPCR SuperMix Plus Kit (Novoprotein, China), and detection was carried out using an Applied Biosystems 7500 Real-Time PCR System (Thermo Fisher Scientific, USA). β-Actin was chosen as an internal reference. The comparative Ct approach was used to calculate the fold changes in relative gene expression (fold change = $2^{-\Delta\Delta Ct}$). The primers used in real-time PCR are listed in S2 Table and were purchased from Sangon Biotech (Shanghai, China).

## Detection of the expression of CASC19 in hepatocarcinoma tissues by RT–qPCR

Tissue cDNA chips containing 14 liver cancer tissues and paired para-carcinoma tissues that were β-actin calibrated were purchased from Shanghai Outdo Biotech (Shanghai, China). RT–qPCR was used to examine the expression of CASC19 using an Applied Biosystems 7500 Real-Time PCR System (Thermo Fisher Scientific, USA). The tissue chips used in our study were commercially available and obtained from Outdo Biotech Company (Shanghai). The application of tissue chips in this research was approved by the Ethics Committee of Shanghai Outdo Biotech Company (YB M-05-01). The study was conducted in accordance with relevant guidelines and regulations, and written informed consent was signed by each participant.

## Colony formation assay

Two days after transfection, Hep3B cells were plated at a density of 5,000 cells/well in 6-well plates. Cells were cultured for 10 days, fixed with 4% paraformaldehyde for 15 minutes, washed 3 times with PBS, and stained with 1% crystal violet for 15 minutes. ImageJ software was used to measure colony formation.

## Wound healing assay

The cells were transfected with short RNA molecules at 70–80% confluence and cultivated for an additional 48 hours. When the cells reached 90% confluency, a scratch was made with a yellow pipette tip, and the wells were rinsed three times with PBS to remove loose cells. The cells were then cultured in serum-free DMEM for 48 hours. Images were captured at 0 h, 24 h and 48 h after wounding with a NEXCOPE NIB900-FL microscope at 200× magnification.

## Migration and invasion assays

For the migration assay, a total of $1 \times 10^5$ transfected Hep3B cells were seeded in the upper chambers of a Transwell (Corning, USA) with 100 μl of serum-free DMEM, and the lower chamber was filled with DMEM with 20% FBS. After culturing for 24 h, no migrated Hep3B cells were wiped off with a cotton swab, and Hep3B cells on the bottom of the chamber were fixed with 4% paraformaldehyde for 15 min and stained with 1% crystal violet for 15 min. Then, the cells were counted and averaged across 5 random fields (× 200 magnification). The invasion assay followed a similar experimental protocol as the migration assay. However, Matrigel was applied to the top chambers (Corning, USA). Each experiment was performed three times.

## Statistical analysis

All statistical analyses were conducted using the R software (R: A Language and Environment for Statistical Computing, R Core Team, R Foundation for Statistical Computing, Vienna, Austria, 2020, https://www.R-project.org) and GraphPad Prism 9. Student's t test or the Wilcoxon test was used to compare continuous data. Continuous variables are expressed as the mean ± standard deviation (SD). To compare continuous variables, unpaired Student's t tests were used. Student's t test or one-way ANOVA was used to assess differences between experimental groups. Cytoscape was used to establish a coexpression network of 6 prognostic m7g-related lncRNAs-mRNAs. Principal component analysis (PCA) was performed to investigate the distribution of HCC patients with prognostic signatures. Cox regression or Kaplan–Meier analysis was used to determine the survival time between groups. ROC analysis was used to assess the sensitivity and specificity. P<0.05 was taken as a criterion of a significant difference. The literature search, data extraction and analysis were performed by two independent researchers to reduce bias in statistics.

# Results

## Detection of differentially expressed m7G-related genes and functional annotation

The flowchart of the whole study is shown in Fig 1. According to the criteria of FDR < 0.05 and |logFC|>1, we identified 20 out of 47 candidate genes that were significantly upregulated in HCC samples (Fig 2A). We further analyzed the interactions among the 20 m7G genes. As illustrated in Fig 2B, TGS1, PABPC1, RAMAC, CDK7, CDK1, GTF2H4, POLR2F, and PSMC4 exhibited weak to moderate associations with other genes. CNB1 and CDK1 exhibited

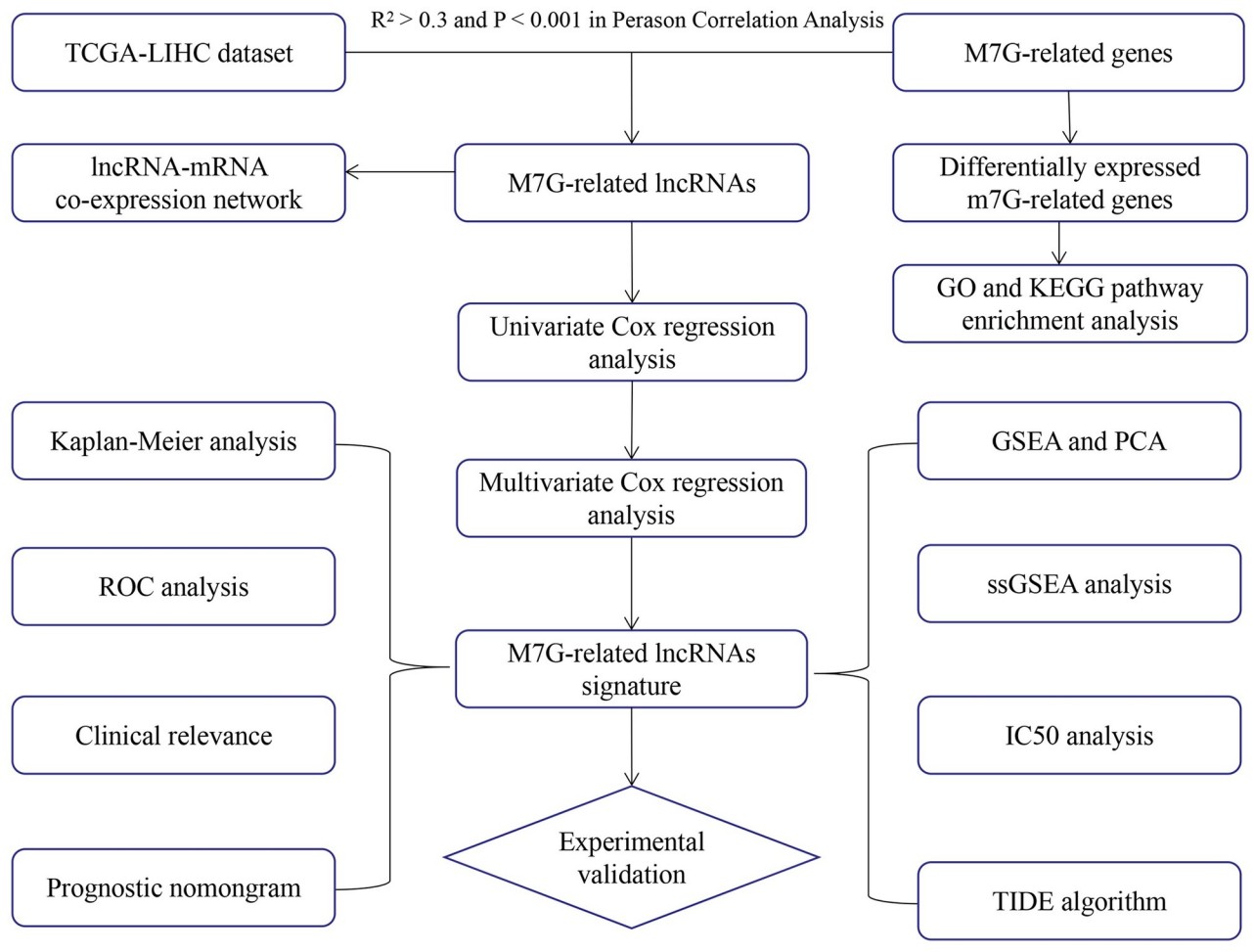

**Fig 1. Flow chart of the study.**

the greatest connection. The expression levels of 20 differentially expressed genes (DEGs) were determined to investigate the basic signaling pathways and GO. KEGG pathway analyses revealed that m7G-related DEGs were primarily enriched in RNA polymerase, basal transcription factors, and nucleotide excision repair. According to GO analysis, the biological processes of DEGs were primarily enriched in mRNA metabolic process, 7-methylguanosine RNA capping and RNA capping. The cellular components of DEGs were mainly enriched in RNA polymerase II, holoenzyme, DNA-directed RNA polymerase complex and nuclear DNA-directed RNA polymerase complex. The molecular functions of DEGs were mainly enriched in RNA polymerase II CTD heptapeptide repeat kinase activity, ATPase activity, and coupled and DNA-dependent ATPase activity (Fig 2C and 2D).

## Detection of the prognostic m7G-related lncRNAs and the lncRNA–mRNA co-expression network

With the criteria of $|R^2| > 0.3$ and P<0.001, we performed Pearson's correlation analysis to examine the co-expression connection between lncRNAs and m7G-related genes. Accordingly, we identified 578 m7G-related lncRNAs based on the TCGA database. Univariate Cox regression analysis showed that 91 m7G-related lncRNAs were highly correlated with HCC prognosis (S3 Table). Multivariate Cox regression analysis revealed 6 m7G-related lncRNAs

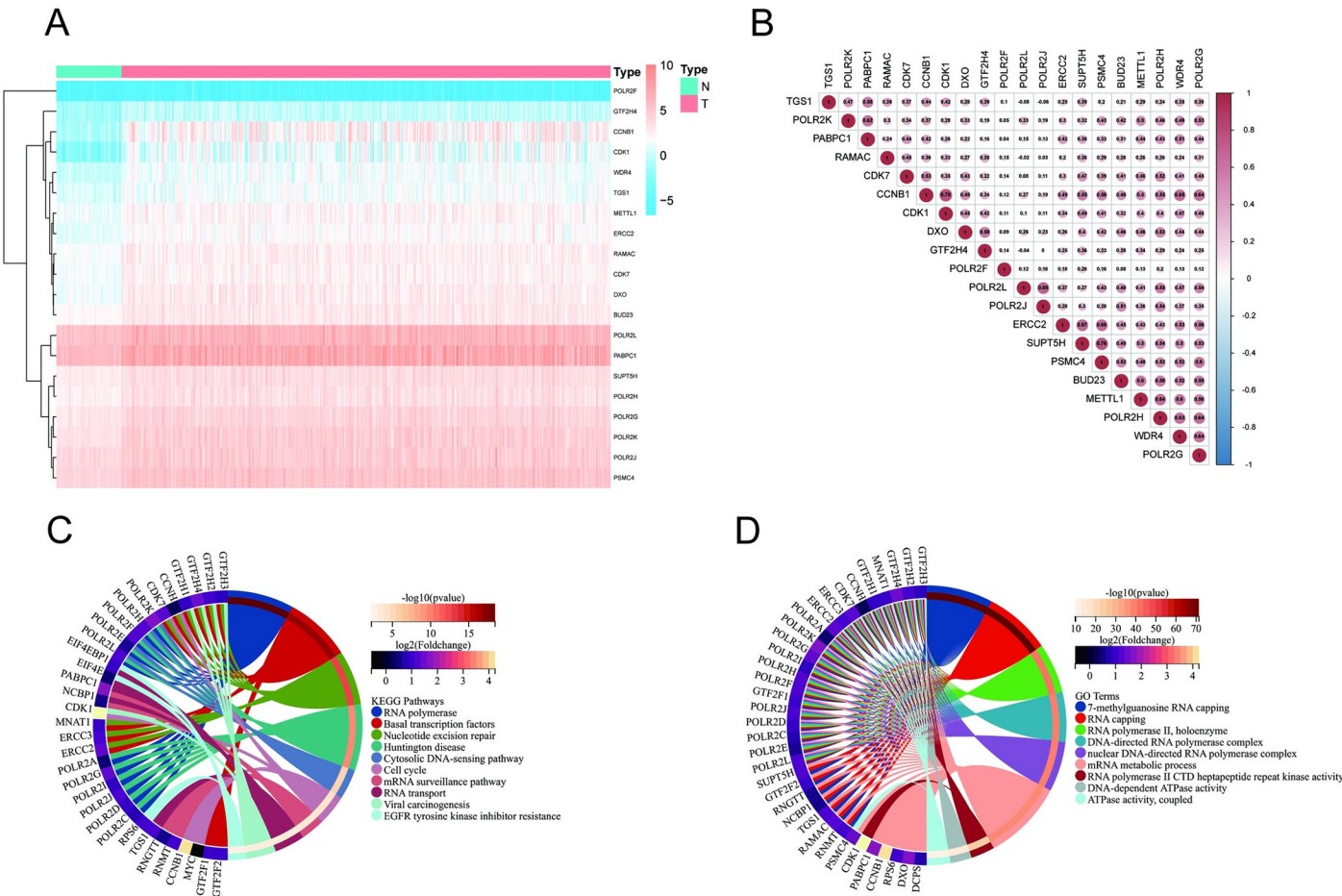

**Fig 2. Differential expression analysis. (A)** Heatmap of 20 differentially expressed m7G-related genes in the tumor and normal groups. **(B)** Pearson's correlation analysis of the m7G regulators. **(C, D)** KEGG and GO enrichment analyses of the m7G-related differentially expressed genes (DEGs).

(AC009403.1, CASC19, AC103760.1, AL117336.3, AC015908.3, AC099850.3) for use in constructing a predictive model (Fig 3A). According to Fig 3B, we further established and visualized a coexpression network of m7G-related lncRNAs-mRNAs with Cytoscape. The link between the 14 mRNAs and the 6 lncRNAs is shown in a Sankey diagram (Fig 3C). AC015908.3 and AC103760.1 were protective factors, and AC009403.1, AC099850.3, AL117336.3 and CASC19 were confirmed to be risk factors.

## Construction of the predictive signature of m7G-related lncRNAs

The prognostic model was built using the 6 m7G-related lncRNAs, and the risk score was obtained using the following formula:

$$
\begin{aligned}
Risk\ score \quad &= (0.706658 \times expression\ levels\ of\ AC009403.1) \\
&+ (0.246827 \times expression\ levels\ of\ CASC19) \\
&+ (-0.35061 \times expression\ levels\ of\ AC103760.1) \\
&+ (0.617499 \times expression\ levels\ of\ AL117336.3) \\
&+ (-0.49976 \times expression\ levels\ of\ AC015908.3) \\
&+ (0.269682 \times expression\ levels\ of\ AC099850.3)
\end{aligned}
$$

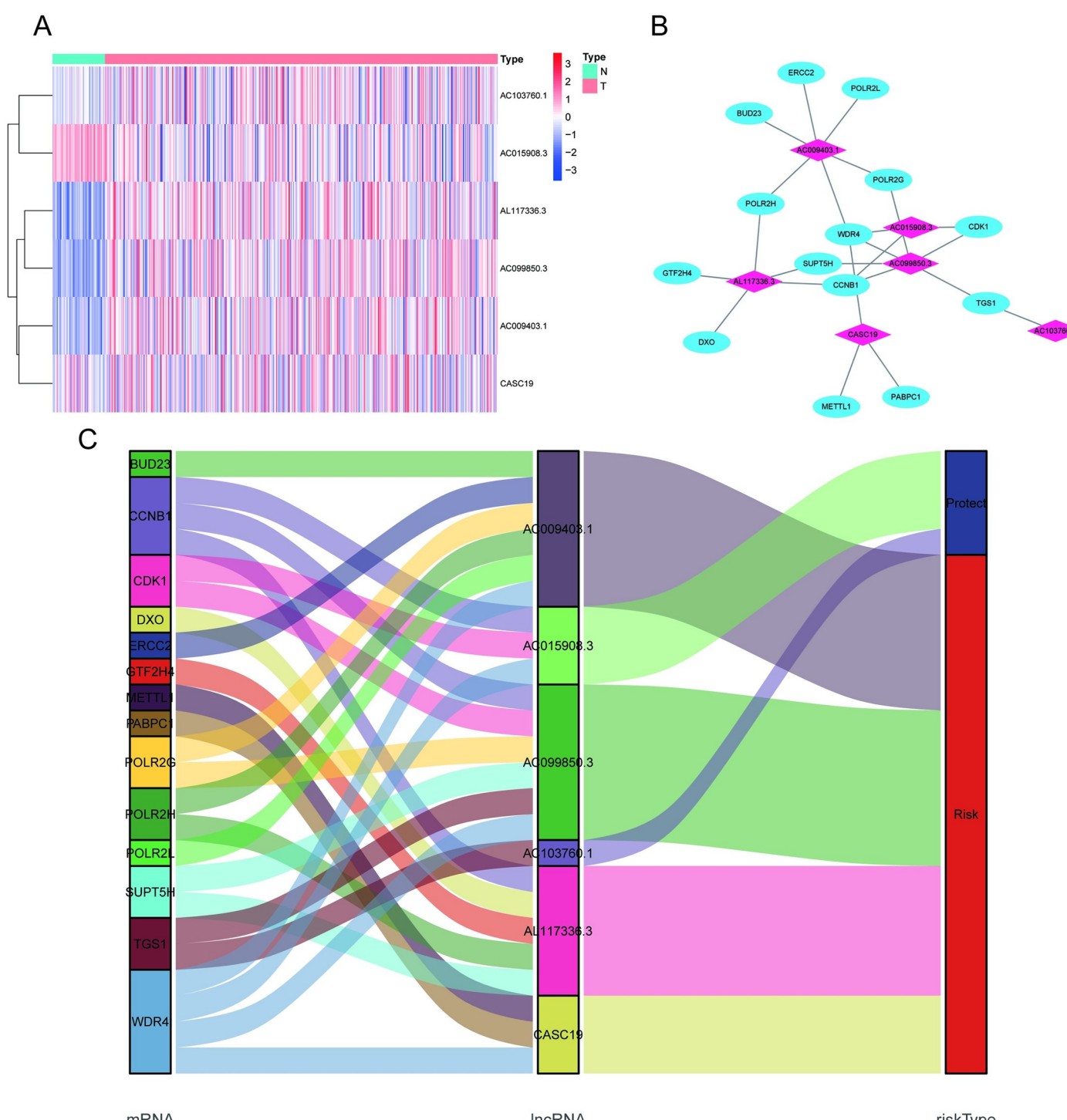

**Fig 3. Expression patterns and lncRNA–mRNA coexpression network of 6 m7G-related lncRNAs with prognostic value. (A)** Heatmap of 6 m7G-related lncRNAs in HCC and normal samples. **(B)** A network of coexpression of 6 m7G-related lncRNAs and mRNAs. **(C)** Sankey diagram of 6 predicted m7G-related lncRNAs and mRNAs.

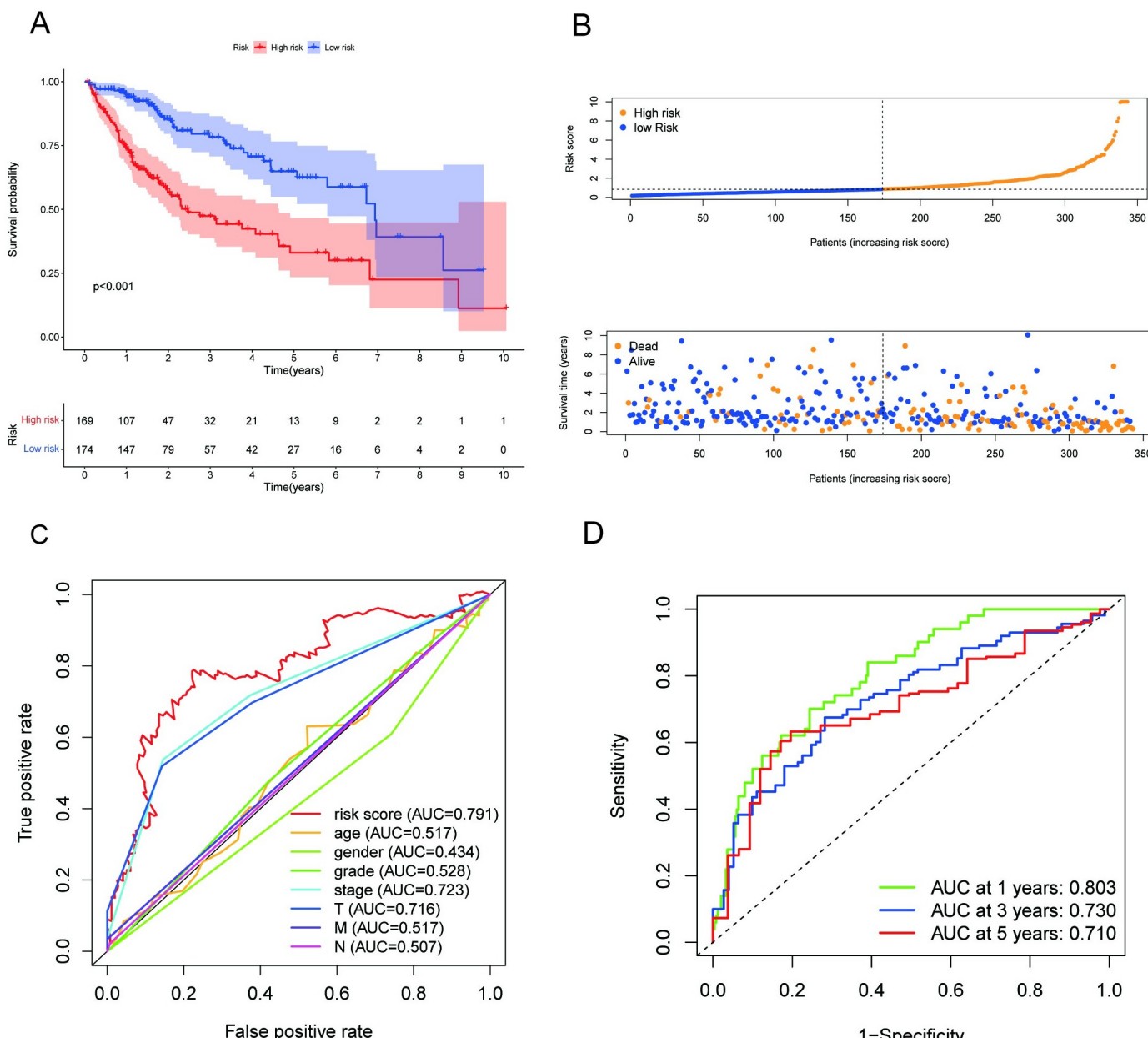

**Fig 4. Relationship between the prognostic signature and HCC patient outcome.** (**A**) Kaplan–Meier curve of overall survival in low-risk and high-risk patients. (**B**) Survival time, life state, and risk score patterns in HCC patients. (**C**) Prognostic signature and clinical and pathological factor ROC curves. (**D**) Prognostic hallmark ROC curves for 1-year, 3-year and 5-year survivability.

The median risk score was used as a cutoff number to separate HCC patients into two groups: low risk (n = 174) and high risk (n = 169). According to the Kaplan–Meier survival analysis, patients in the high-risk group exhibited a considerably lower OS than those in the low-risk group (Fig 4A, $p < 0.001$). We assessed the prognosis index and survival state of HCC patients in various groups, as displayed in Fig 4B, and patients in the high-risk group had a higher risk of death than patients in the low-risk group. According to the ROC curve study, the AUC of the m7G-related lncRNA predictive signature was 0.791, which was superior to

that of other clinicopathological variables (Fig 4C). Time-dependent ROC plots were also used to assess diagnostic efficiency, with AUCs of 0.803, 0.730, and 0.710 for 1-year, 3-year and 5-year survival, respectively (Fig 4D). According to these findings, the 6 m7G-related lncRNA signature exhibited a remarkable ability to predict HCC prognosis.

## Development and validation of the nomogram

We created a nomogram to estimate the 1-year, 3-year and 5-year OS of HCC patients based on the risk score and clinical characteristics to learn more about the predictive performance of the 6 m7G-related lncRNA signature (Fig 5A). Predictive nomogram models were further constructed to assess the utility of the risk score as a means of predicting the 1-, 3-, and 5-year

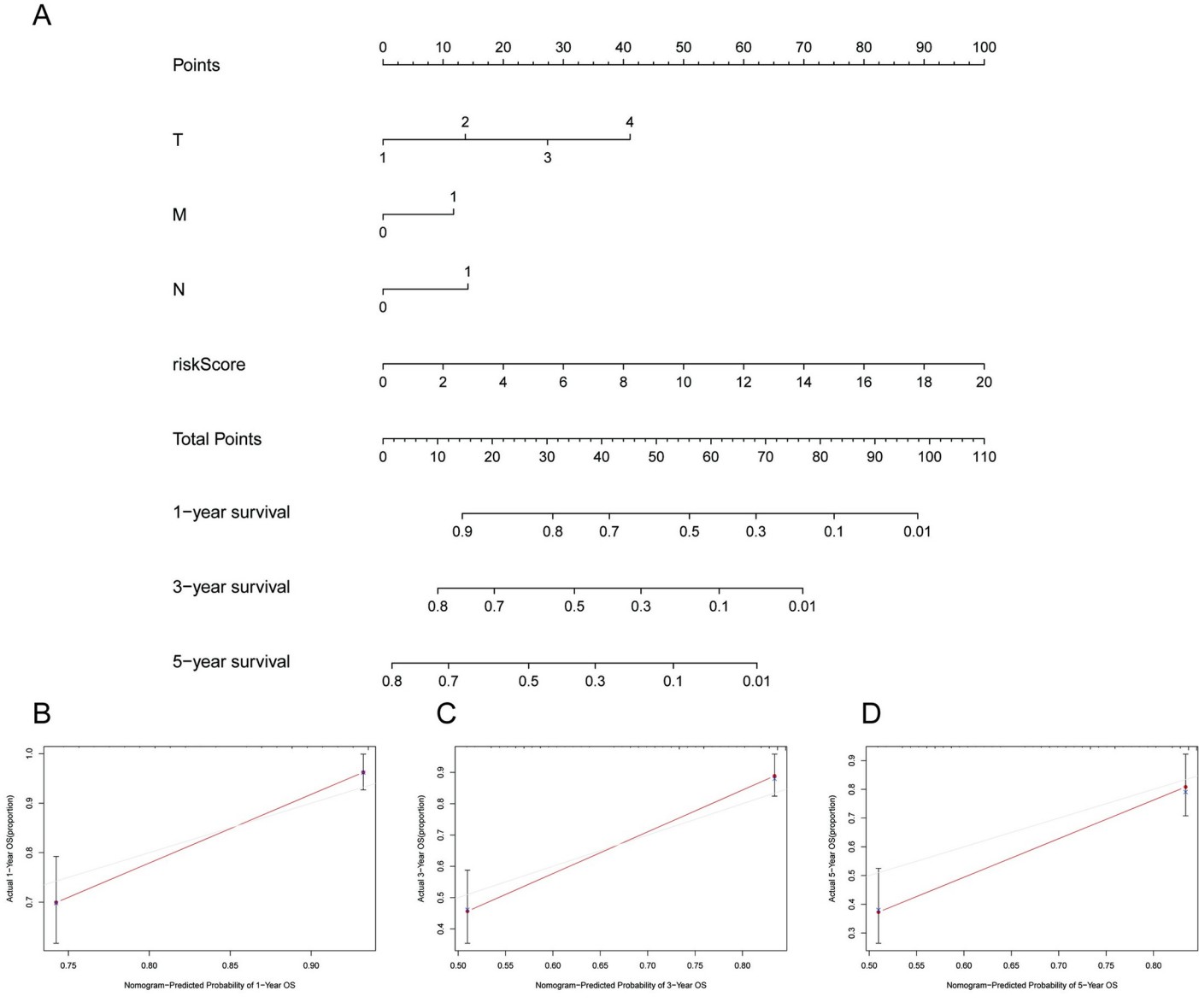

**Fig 5. Development and validation of the nomogram. (A)** Prediction of the nomogram based on clinical variables and risk score. **(B-D)** Calibration curves for predictive OS.

prognosis of HCC patients, with calibration curves demonstrating that these risk scores offered good prognostic utility for all three of these time intervals. As illustrated in Fig 5B–5D, the calibration curves showed high agreement between the observed and expected OS results at 1 year, 3 years and 5 years.

## Correlations between risk scores and clinical variables

We further investigated the association between the risk score and each clinical characteristic. The heatmap illustrated the clinicopathological characteristics and expression levels of the six lncRNAs in the high- and low-risk groups (Fig 6A). The related scatter diagrams in Fig 6B–6D

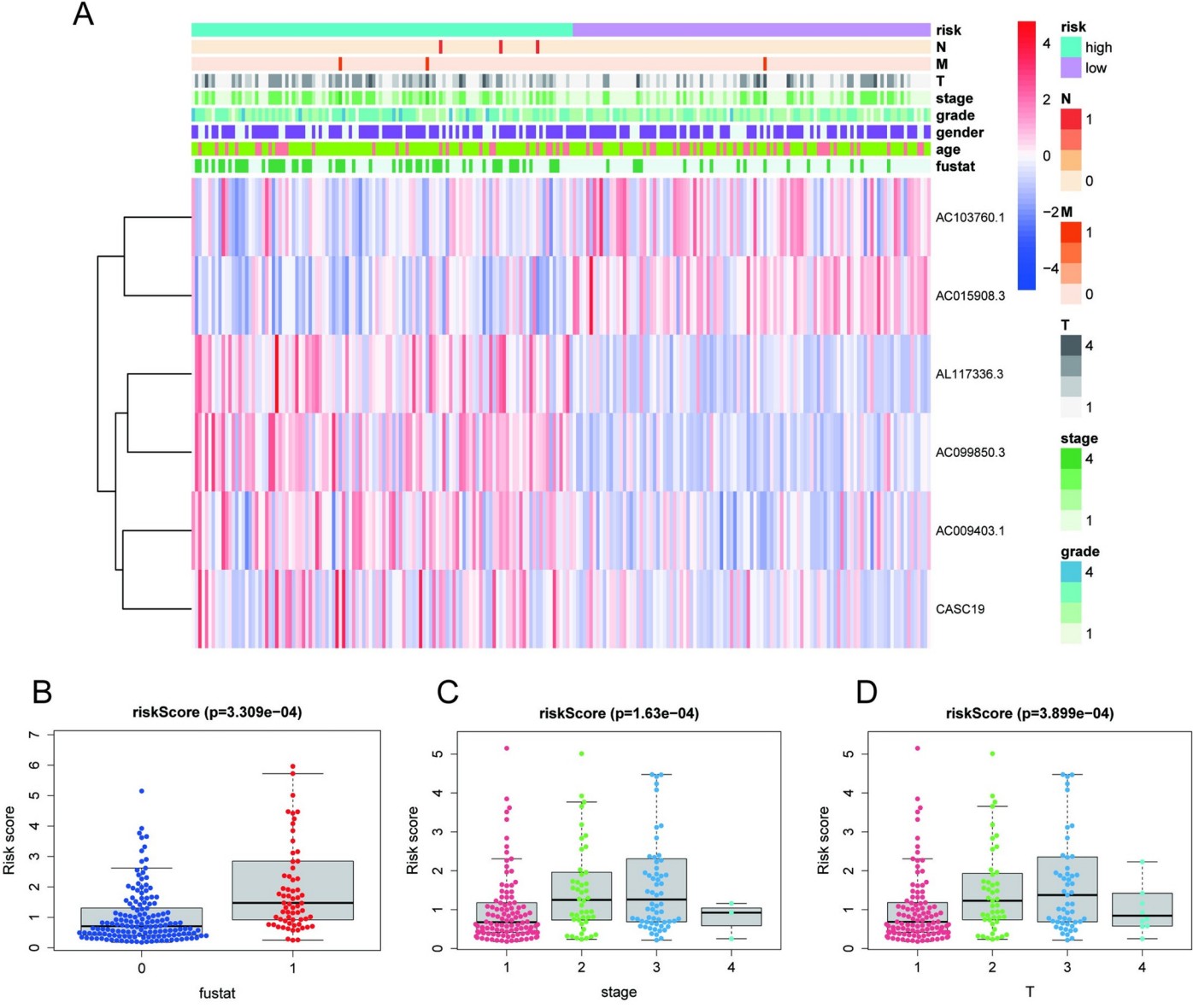

**Fig 6. The expression of the six m7G-associated lncRNAs and clinical variables were shown to be connected. (A)** Heatmap of six prognostic m7G-related lncRNAs and clinical variables based on subgroups. **(B-D)** Risk scores for survival status, clinical stage, and T stage.

reveal that the risk score was linked with survival state (p < 0.001), clinical stage (p < 0.001), and T stage (p < 0.001).

## Internal validation of the predictive signature

We further evaluated the robustness of the predictive signature by internal validation. First, the 343 HCC patients were randomly separated into two cohorts (internal cohort 1, n = 172; internal cohort 2, n = 171). Next, we calculated a risk score in the internal validation cohorts using the same formula. As expected, the results of our internal validation were highly consistent with the results of the entire dataset. According to Fig 7, the OS rates of the high-risk

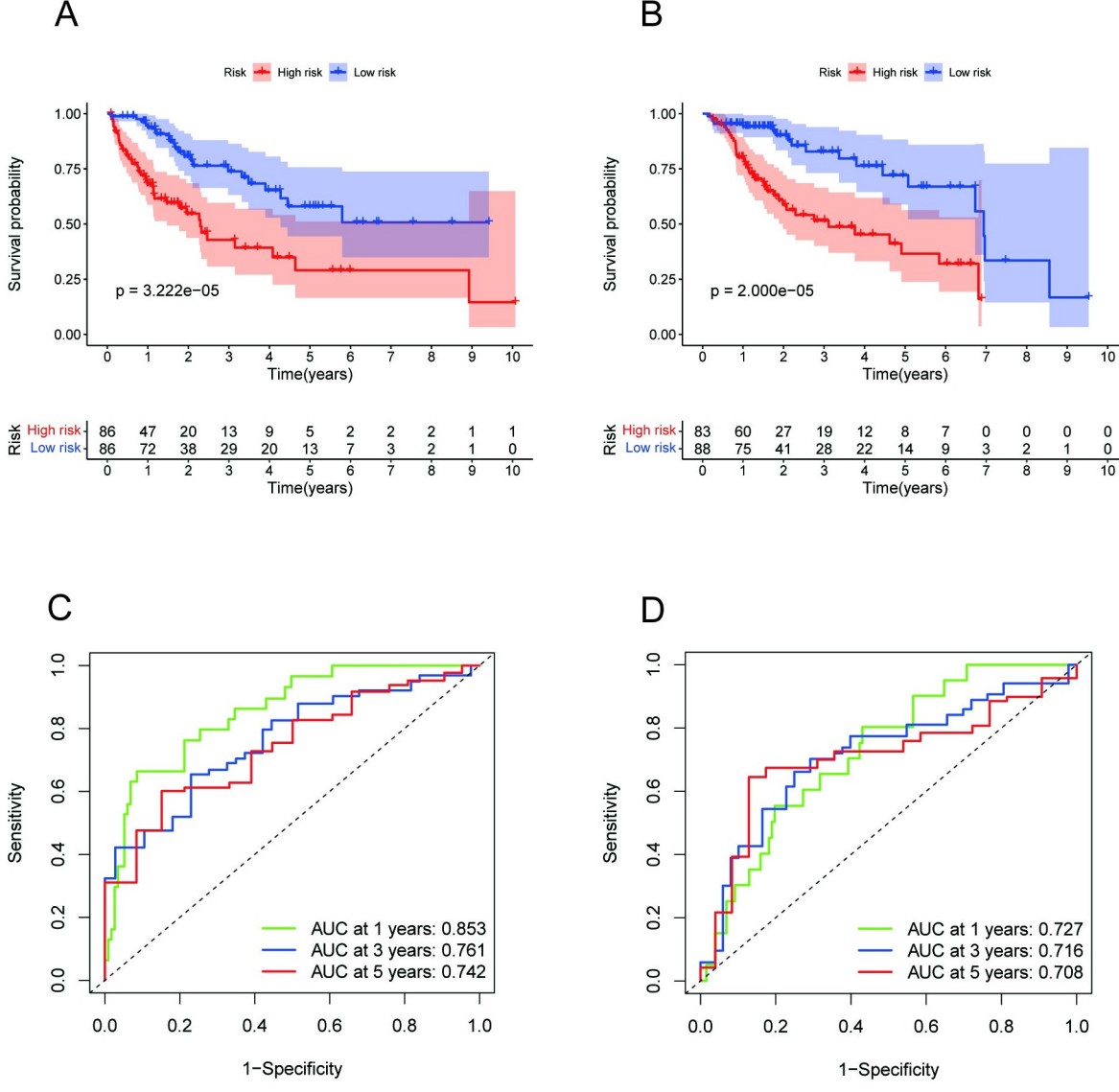

**Fig 7. Validation based on internal cohorts. (A)** Kaplan–Meier analysis of internal cohort 1. **(B)** Kaplan–Meier analysis of internal cohort 2. **(C)** ROC curves of 1-year, 3-year and 5-year survival in internal cohort 1. **(D)** ROC curves of 1-year, 3-year and 5-year survival in internal cohort 2.

patients were markedly shorter than those of the low-risk patients in our internal cohorts, and the ROC curves of the two cohorts presented good predictive value for survival rates.

## Gene set enrichment analysis and principal component analysis

Based on the whole genome, m7G-related gene sets, m7G-related lncRNAs, and the m7G-related lncRNA prognostic signature, we created PCA charts to depict the distribution patterns of HCC patients. The results of our analysis showed that compared with the other 3 methods, the m7G-related lncRNA prognostic signature presented a better performance in distribution (Fig 8A–8D). We also used GSEA to identify how 6 m7G-related lncRNAs behaved at the molecular level. As shown in S4 Table, 86 enrichment pathways with significant variations between the low- and high-risk groups were identified based on an FDR $< 0.25$ and $p < 0.05$. The top five signaling pathways in the high-risk category were RNA degradation, pyrimidine metabolism, base excision repair, oocyte meiosis, and cell cycle. The top five signaling pathways in the low-risk group were complement and coagulation cascades, fatty acid metabolism, retinol metabolism, primary bile acid biosynthesis, and drug metabolism cytochrome p450 (Fig 8E and 8F). According to the findings of our study, high-risk patients were closely related to RNA metabolism pathways.

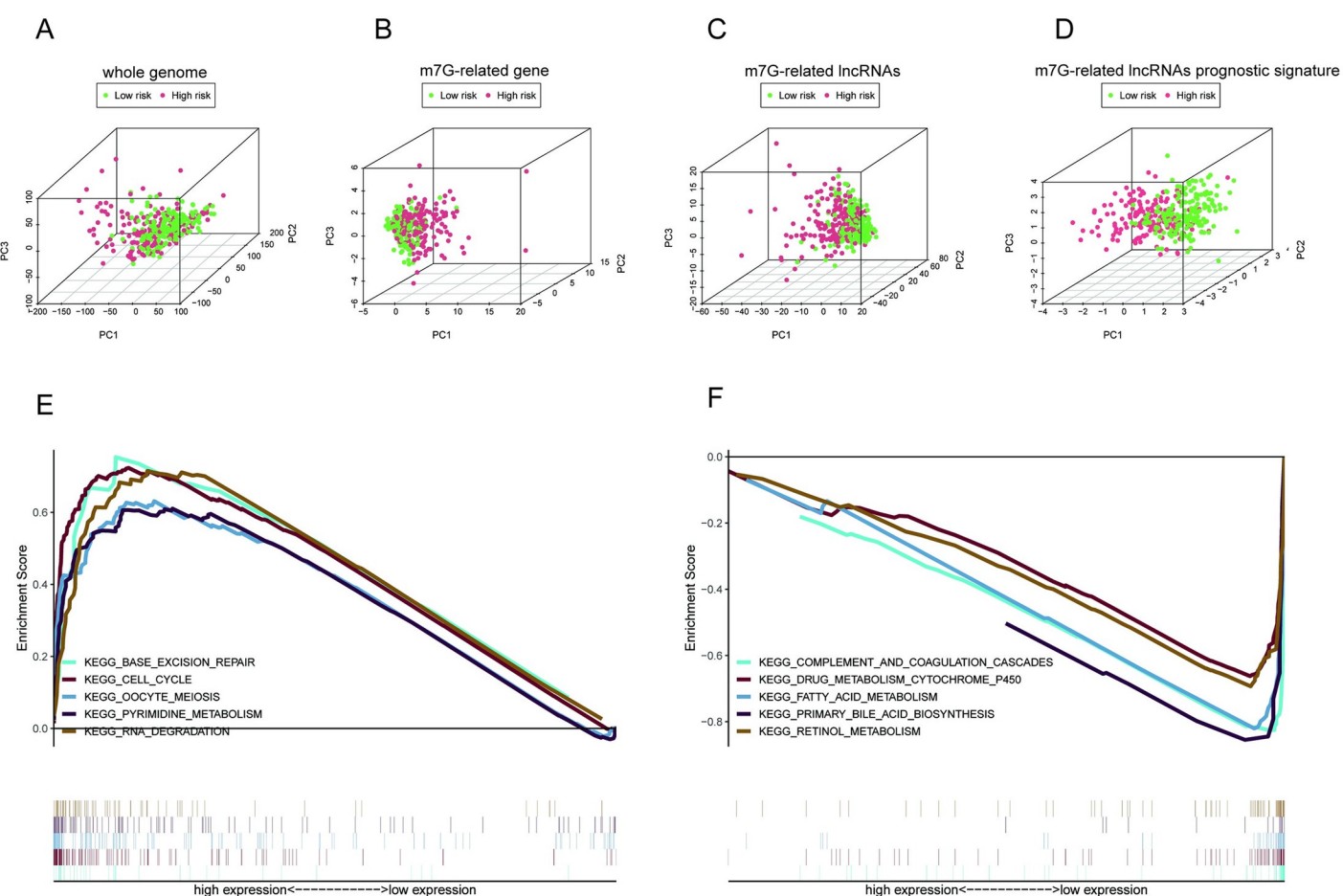

**Fig 8. Signaling analyses of the low-risk and high-risk groups.** PCA charts depict patient distribution based on **(A)** the whole genome; **(B)** m7G-related genes; **(C)** m7G-related lncRNAs; and **(D)** the m7G-related lncRNA prognostic signature. **(E, F)** The top five enriched signaling pathways in the subgroups.

## Correlations between the risk scores and immunological infiltration and immune function

To better investigate the relationship between risk scores and immunological infiltration and associated functions, ssGSEA was used to calculate the enrichment scores of various immune cells. According to Fig 9A and 9B cells, macrophages, mast cells, neutrophils, NK cells, and TILs were substantially different in the high- and low-risk groups. We also found substantial variations in immune function in terms of APC coinhibition, cytolytic activity, MHC class I, parainflammation, type I IFN response, and type II IFN response (Fig 9B). In summary, the above results indicated that m7G-related lncRNAs were related to immunological infiltration and immune function in HCC patients.

## Prediction of response to immunotherapy and identification of potential chemotherapy drugs

Given the significance of checkpoint treatment, we further investigated the variations in immune checkpoint expression between subgroups. According to Fig 10A, except for D40LG and TNFSF14, the expression levels of other immune checkpoints were considerably higher in the high-risk group than in the low-risk group, which may present new insights into immune checkpoint therapy of HCC. The TIDE algorithm was used to further assess the potential efficacy of immunotherapy in different subgroups. In our results, the high-risk group presented a lower TIDE score than the low-risk group, which suggested that the patients in the high-risk group would be more responsive to immune therapy (Fig 10B). We also found that T-cell exclusion scores were higher in the high-risk group, while T-cell dysfunction scores were lower in the low-risk group (Fig 10C and 10D). Moreover, we investigated the relationship between the risk score and the effectiveness of conventional chemotherapy in the treatment of HCC (Fig 10E–10K). The IC50 values for bleomycin, doxorubicin, gemcitabine, and lenalidomide were relatively low in the high-risk group, while the IC50 values for dasatinib, erlotinib,

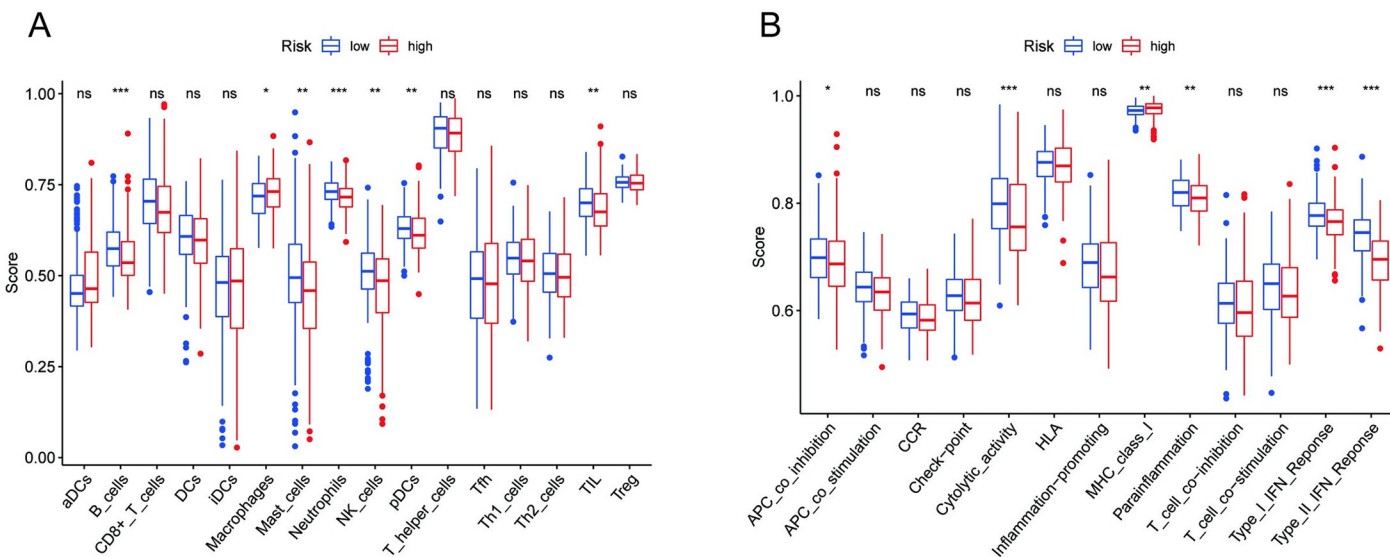

**Fig 9. The risk scores of immunological infiltration and immune functions in the subgroups.** (A) The ssGSEA scores of immunological infiltration. (B) The ssGSEA scores of immune functions.

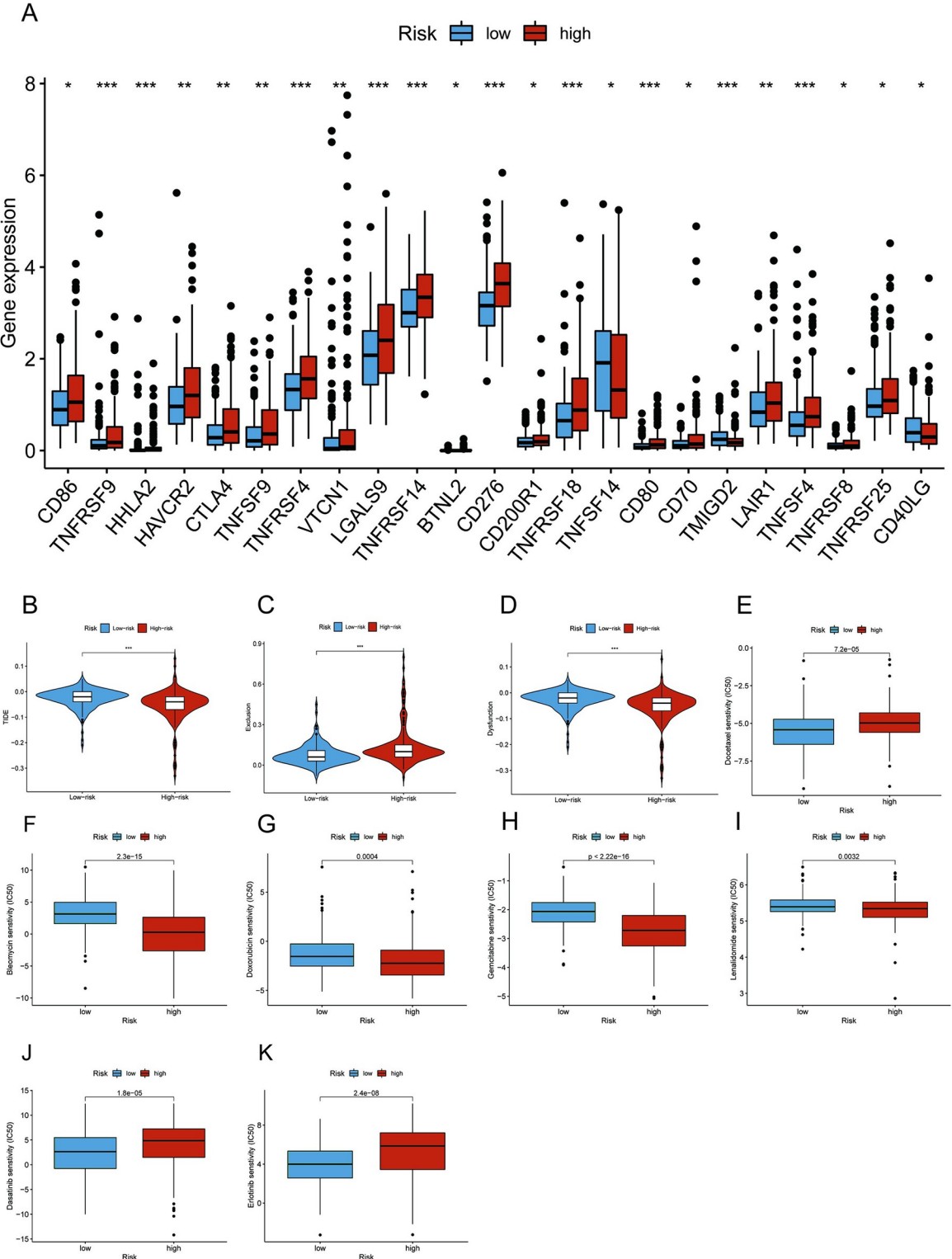

**Fig 10. Immune checkpoint expression and chemotherapy drug sensitivity in the subgroups. (A)** Expression patterns of immune checkpoints. **(B-D)** TIDE, T-cell exclusion and T-cell dysfunction scores in different subgroups. **(E-K)** IC50 of bleomycin, doxorubicin, gemcitabine, lenalidomide, dasatinib, erlotinib, and docetaxel.

and docetaxel were relatively high in the high-risk group, which was conducive to the precise treatment and individualized treatment of high-risk patients.

## Knockdown of CASC19 inhibited proliferation, migration and invasion of hepatocellular carcinoma cells

CASC19 has been documented to be associated with the development and progression of a number of malignancies, including gastric cancer, glioma, and pancreatic cancer [32, 33], but it has not been reported in HCC. Considering that the risk model-related lncRNAs were closely related to the prognosis of HCC, we selected CASC19 for further analysis. As shown in Fig 11A and 11B, CASC19 was highly expressed in HCC in the TCGA data, which was confirmed by qPCR results in 14 liver cancer tissues and paired para-carcinoma tissues (The average expression level: 3.945 to 1.000, P <0.05). Knockdown of CASC19 was also conducted in Hep3B cells to explore its possible functions in HCC. As shown in Fig 11C, siCASC19-2 and siCASC19-139 significantly reduced endogenous CASC19 expression at 48 hours posttransfection and were selected for subsequent experiments. Colony formation assays revealed that knockdown of CASC19 remarkably suppressed cell viability in Hep3B cells (The average cell clone numbers for si-NC, siRNA-2, and siRNA-139 were 1535, 985, 935, P <0.05) (Fig 11D). Transwell assays demonstrated that knockdown of CASC19 notably reduced cell migration and invasion (For the migration and invasion assays, the average cell clone numbers of si-NC, siCASC19-2, and siCASC19-139 came to 4293, 2495, 2901 and 179, 54, 51, respectively, P <0.05) (Fig 11E and 11F). Additionally, the results of wound healing studies showed that cells with CASC19 knockdown had decreased migratory ability (For wound healing experiments, the average migration rates of si-NC, siRNA-2, and siRNA-139 at 24 and 48 hours were 15.43%, 7.21%, 5.16% and 20.8%, 9.17%, 8.47%, respectively, P <0.05) (Fig 11G).

## Discussion

HCC is a highly heterogeneous disease, and survival times vary widely among patients with similar clinical information. Thus, finding effective strategies for early detection, prognostic analysis, and individualized therapy of HCC is critical. A growing number of studies have established that posttranscriptional RNA modifications play a significant role in modulating gene expression as well as carcinogenesis and development; the most common modifications are m6A, m5C, m1A, and m7G [34, 35]. WDR4 and METTL1 have been implicated in hepatocarcinogenesis and progression in recent studies [16]. In our study, we analyzed the correlation and functional enrichment analysis of 20 m7G-related DEGs in HCC patients based on the TCGA database.

Although lncRNAs not translated, they are abundantly expressed and closely associated with various cancers, and their aberrant transcription and alterations are strongly linked to carcinogenesis, invasion, and tumor stage. Xu Z. *et al.* constructed a lncRNA signature to predict prognosis of and immunotherapy effects on HCC [36]. Recently, scientists have concentrated on identifying RNA modification-related lncRNA signatures to assess the outcomes of cancer patients [37–42]. Li *et al.* developed a risk model based on m6A-related lncRNAs that can be used to evaluate treatment effects and outcomes in HCC [43]. However, there have been no reports on the prediction of the prognosis of HCC by constructing m7G-related lncRNA risk signatures. To the best of our knowledge, this study is the first comprehensive investigation of the roles of m7G-related lncRNAs in HCC.

In this work, 6 m7G-related lncRNAs were recognized and further applied to construct a predictive signature. In addition, we found that 14 mRNAs were significantly co-expressed with lncRNAs. METTL1 and WDR4 are two important regulators in the process of m7G

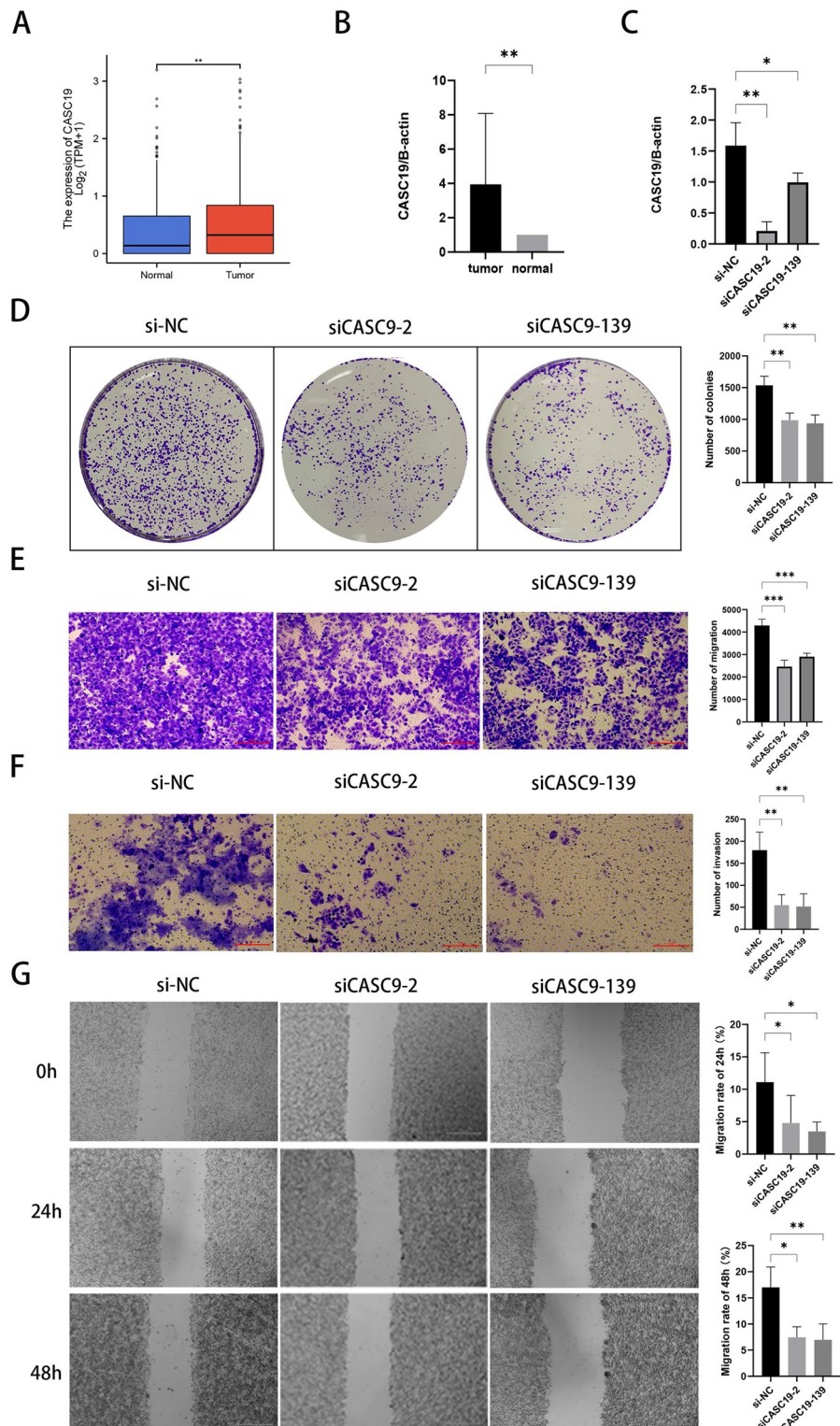

**Fig 11. Role of CASC19 in cell proliferation, migration, and invasion in HCC cells. (A)** CASC19 expression in the TCGA database. **(B)** RT–qPCR results of CASC19 expression in tissue samples. **(C)** Transfection efficiency of siCASC19 in Hep3B cells. **(D)** Colony formation assays were performed to measure the proliferation of Hep3B cells after CASC19 knockdown. **(E-G)** Transwell assays and wound healing assays were applied to detect the migration (E, G) and invasion (F) of Hep3B cells after CASC19 downregulation. * p < 0.05, ** p < 0.01, *** p < 0.001.

methylation. Xia P. *et al*. observed that upregulation of WDR4 expression increases the methylation level of m7G in HCC and promotes HCC cell proliferation [15]. METTL1-mediated m7G modification of Arg-TCT tRNA has also been reported to promote oncogenic transformation [44]. CASC19, AL117336.3, AC015908.3 and AC099850.3 were indicated by different bioinformatics analyses to have prognostic value and were applied to build risk prediction models [45–48]. In recent experiments, overexpression of CASC19 enhanced the invasive and migratory functions of colorectal and gastric cancer cells, while knockdown of CASC19 inhibited proliferation and migration in vitro [32, 49]. Interestingly, CASC19 siRNA (siCASC19) was observed to inhibit autophagy and promote apoptosis in nasopharyngeal carcinoma [50]. We carried out an in vitro preliminary experimental validation given the numerous functions of CASC19 in cancers. The results indicated that CASC19 was highly expressed in liver cancer tissues and that downregulation of CASC19 inhibited the proliferation, invasion and migration of Hep3B cells in vitro (Fig 11E–11G). Additionally, AC099850.3 was reported to promote the migration and proliferation of HCC cells [51]. An increasing number of studies have demonstrated the value of lncRNAs in tumor diagnosis and prognosis and in understanding the molecular mechanisms underlying cancers [52–54].

Furthermore, GSEA revealed that RNA degradation, pyrimidine metabolism, base excision repair, oocyte meiosis and the cell cycle were mainly enriched in the high-risk group. In prostate cancer, Li J. *et al*. reported that YTHDF2 might regulate tumor suppressor mRNA degradation to promote AKT phosphorylation in an m6A-dependent mechanism [55]. The tumor immune microenvironment is crucial in the initiation and progression of HCC. The ssGSEA results showed that macrophages presented higher scores in the high-risk group. This discovery is consistent with prior research showing that increased tumor-related macrophage infiltration results in a poor prognosis in gastric and esophageal cancer [56, 57]. Tumor-associated macrophage-induced immune responses are already considered critical determinants of tumor progression [58]. Tumor-associated macrophages can also perform pretumor activities, such as enhancing tumor cell proliferation, invasion, angiogenesis, and extracellular mesenchymal remodeling and inhibiting antitumor immune surveillance [59].

Given that immunotherapy has proven to present enormous potential in the cancer treatment of HCC patients, the identification of patients who can benefit most from immunotherapy is crucial and needed [60, 61]. Therefore, exploring the molecular mechanisms of m7G-related mRNAs and lncRNAs that regulate immune escape and immunosuppression may lead to new immunotherapy approaches. TIDE is an innovative computational algorithm recognized as a highly reliable method for predicting the clinical response of patients to immunotherapy [31]. According to our findings, low-risk patients had higher TIDE and T-cell dysfunction scores than high-risk patients, suggesting that the lower immunotherapy response might be related to immune evasion and T-cell dysfunction. Additionally, patients in the high-risk group seemed to be more responsive to bleomycin, doxorubicin, gemcitabine, and lenalidomide. Thus, our study might provide a strategy for optimizing the regimens of chemotherapy and immunotherapy for patients in different groups, thereby increasing the response rate and improving the prognosis of HCC patients.

Of course, there are several limitations in the current study. First, although the method of identifying m7G-associated lncRNAs by co-expression analysis is the most common, it is not sufficiently accurate. Second, due to the lack of suitable datasets, we did not use external datasets containing lncRNA expression profiles to verify the generalizability of the m7G-related lncRNA signature. Third, we have only conducted preliminary experiments, and further animal experiments should be carried out to explore the molecular roles of key lncRNAs in liver cancer.

## Conclusion

In summary, we found that the m7G-related lncRNA signature is an independent predictor of OS in HCC patients. Moreover, our research might provide new insights into prognostic prediction and response to chemotherapy and immunotherapy in HCC based on m7G-related lncRNAs.

## Supporting information

**S1 Table. A total of 47 m7G-related genes.**
(XLSX)

**S2 Table. The primers used in real-time PCR.**
(XLS)

**S3 Table. List of 91 m7G-related lncRNAs.**
(XLSX)

**S4 Table. GSEA analysis.**
(XLSX)

## Acknowledgments

Thanks to the Provincial Key Laboratory of Nephrology of Shanxi Provincial People's Hospital for providing the experimental platform.

## Author Contributions

**Conceptualization:** Li-Li Guo, Mu-Ye Li, Hui Liao, Rong-Shan Li.

**Data curation:** Yue-Ling Peng, Ya-Fang Dong, Li-Li Guo, Mu-Ye Li.

**Formal analysis:** Yue-Ling Peng, Ya-Fang Dong, Mu-Ye Li.

**Funding acquisition:** Rong-Shan Li.

**Investigation:** Li-Li Guo.

**Methodology:** Yue-Ling Peng, Li-Li Guo, Mu-Ye Li, Hui Liao, Rong-Shan Li.

**Resources:** Rong-Shan Li.

**Software:** Mu-Ye Li.

**Supervision:** Ya-Fang Dong, Li-Li Guo.

**Validation:** Yue-Ling Peng, Ya-Fang Dong, Mu-Ye Li.

**Visualization:** Yue-Ling Peng.

**Writing – original draft:** Yue-Ling Peng, Mu-Ye Li, Hui Liao.

**Writing – review & editing:** Li-Li Guo, Rong-Shan Li.

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
