## [Decision Letter · Decision Letter 0]

26 Jun 2023

PONE-D-23-06969Identification and validation of a m7G-related lncRNA signature for predicting the prognosis and therapy response in hepatocellular carcinomaPLOS ONE

Dear Dr. Shan-Li,

Thank you for submitting your manuscript to PLOS ONE. After careful consideration, we feel that it has merit but does not fully meet PLOS ONE’s publication criteria as it currently stands. Therefore, we invite you to submit a revised version of the manuscript that addresses the points raised during the review process.

Please submit your revised manuscript by 10th July 2023. If you will need more time than this to complete your revisions, please reply to this message or contact the journal office at plosone@plos.org. Please include the following items when submitting your revised manuscript:A rebuttal letter that responds to each point raised by the academic editor and reviewer(s). You should upload this letter as a separate file labeled 'Response to Reviewers'.A marked-up copy of your manuscript that highlights changes made to the original version. You should upload this as a separate file labeled 'Revised Manuscript with Track Changes'.An unmarked version of your revised paper without tracked changes. You should upload this as a separate file labeled 'Manuscript'.If applicable, we recommend that you deposit your laboratory protocols in protocols.io to enhance the reproducibility of your results. Protocols.io assigns your protocol its own identifier (DOI) so that it can be cited independently in the future. For instructions see: https://journals.plos.org/plosone/s/submission-guidelines#loc-laboratory-protocols. Additionally, PLOS ONE offers an option for publishing peer-reviewed Lab Protocol articles, which describe protocols hosted on protocols.io. Read more information on sharing protocols at https://plos.org/protocols?utm_medium=editorial-email&utm_source=authorletters&utm_campaign=protocols.

We look forward to receiving your revised manuscript.

Kind regards,

Afsheen Raza, PhD

Academic Editor

PLOS ONE

Journal Requirements:

“This study was supported by the Natural Science Foundation of Shanxi Province (201903D421020) and the Special Fund of Central Government Guided Local Scientific Development (YDZJSX2021O027).The funders had no role in study design, data collection and analysis, decision to publish, or preparation of the manuscript.”

Reviewers' comments:

Reviewer's Responses to Questions

**Comments to the Author**

1. Is the manuscript technically sound, and do the data support the conclusions?

Reviewer #1: Yes

Reviewer #2: Yes

2. Has the statistical analysis been performed appropriately and rigorously? 

Reviewer #1: Yes

Reviewer #2: Yes

3. Have the authors made all data underlying the findings in their manuscript fully available?

Reviewer #1: Yes

Reviewer #2: Yes

4. Is the manuscript presented in an intelligible fashion and written in standard English?

Reviewer #1: Yes

Reviewer #2: Yes

5. Review Comments to the Author

Reviewer #1: I would like to say that there is not obvious error or problem for this work, although it lacks novelty in scientific sense and potential clinical application. Therefore, it is quite difficult to judge whether to accept it based on the quality of the work or novelty of it.

Also, it is quite hard to conduct further analysis or experiment. Most significant analysis has been done in this work and there is no need for further experiments. Plus, I do not think there will be a high possibility to test RNA m7G to predict prognosis of cancer pt in clinical.

My impression and suggestion is that, to make more scientific sense, I think there can be an explanation on the linkage of the m7G and the related lncRNA expression/function in the signature. We know that the lncRNA is often little in expression and its function is linked to interaction with genome or RNA or protein. Also, a further analysis among the m7G or lncRNAs might be needed. Foe example, what are they in common, structure or function or both? Why are they with prognostic value as a whole but not so much individually?

Reviewer #2: 1. How to identify a total of 47 m7G-related genes in GeneCards ?

2. Is there some references about "|R2| > 0.3 and P <0.001 were considered to indicate m7G-related lncRNAs" in materials and methods? Supply more details about indentification

of M7G-related lncRNAs.

3. Supply more details about "Development and validation of the nomogram" in materials and methods

4. Is an FDR of less than 0.25 considered significance in materials and methods ? Supply more details about "Gene set enrichment analysis"

5. Supply more details about "Prediction of response to immunotherapy" in materials and methods

6. Some minor mistakes need to be noted, such as "CO2" , "2 -△△Ct" , "As illustrated in Fig.B-D"

6. PLOS authors have the option to publish the peer review history of their article (what does this mean?). If published, this will include your full peer review and any attached files.

Reviewer #1: **Yes: **Bingjie Li

Reviewer #2: No

---

## [Author Response · Author response to Decision Letter 0]

28 Jun 2023

Dear Editor and Reviewers:

Thank you for giving us the opportunity to submit a revised draft of the manuscript "Identification and validation of a m7G-related lncRNA signature for predicting the prognosis and therapy response in hepatocellular carcinoma"(PONE-D-23-06969R1) to PLOS ONE. 

We appreciate the time and effort that you dedicated to our manuscript, and we are grateful for your insightful comments and suggestions. We have taken all these comments and suggestions into account, and we have made major corrections in this revised manuscript.

Response to Academic Editor:

Q1. Please ensure that your manuscript meets PLOS ONE's style requirements, including those for file naming. The PLOS ONE style templates can be found at

Reply:Thank you for reminding us of the PLOS ONE style requirements. We understand the importance of adhering to these guidelines and we have taken the necessary steps to ensure our manuscript and files comply with all of PLOS ONE's style requirements, including those for file naming.We hope that these revisions meet your approval and we look forward to hearing from you soon.

Q2. Thank you for stating in your Funding Statement:

“This study was supported by the Natural Science Foundation of Shanxi Province (201903D421020) and the Special Fund of Central Government Guided Local Scientific Development (YDZJSX2021O027).The funders had no role in study design, data collection and analysis, decision to publish, or preparation of the manuscript.”

Reply:Thank you for your comments. We would like to amend the funding statement as follows:

"This study was supported by the Natural Science Foundation of Shanxi Province (201903D421020) and the Special Fund of Central Government Guided Local Scientific Development (YDZJSX2021O027). There was no additional external funding received for this study. The funders had no role in study design, data collection and analysis, decision to publish, or preparation of the manuscript."

We also include the amended Funding Statement within the cover letter.

Q3. Please review your reference list to ensure that it is complete and correct. If you have cited papers that have been retracted, please include the rationale for doing so in the manuscript text, or remove these references and replace them with relevant current references. Any changes to the reference list should be mentioned in the rebuttal letter that accompanies your revised manuscript. If you need to cite a retracted article, indicate the article’s retracted status in the References list and also include a citation and full reference for the retraction notice.

Reply:Thank you for your comment on ensuring the completeness and correctness of our reference list. Upon receiving your suggestions, we have carefully reviewed our citations and the reference list. We found that some of the previously included articles have been retracted, and we appreciate your vigilance in maintaining the integrity of scientific literature.

We have now removed these retracted articles from our reference list. We appreciate your thorough review and thank you for your guidance in this matter.

Response to Reviewer #1: 

I would like to say that there is not obvious error or problem for this work, although it lacks novelty in scientific sense and potential clinical application. Therefore, it is quite difficult to judge whether to accept it based on the quality of the work or novelty of it.

Also, it is quite hard to conduct further analysis or experiment. Most significant analysis has been done in this work and there is no need for further experiments. Plus, I do not think there will be a high possibility to test RNA m7G to predict prognosis of cancer pt in clinical.

My impression and suggestion is that, to make more scientific sense, I think there can be an explanation on the linkage of the m7G and the related lncRNA expression/function in the signature. We know that the lncRNA is often little in expression and its function is linked to interaction with genome or RNA or protein. Also, a further analysis among the m7G or lncRNAs might be needed. Foe example, what are they in common, structure or function or both? Why are they with prognostic value as a whole but not so much individually?

Reply:We would like to express our sincere gratitude for your thoughtful comments and insights. Your acknowledgement of the comprehensive analysis performed in our study is greatly appreciated, and we take to heart your comments on its potential limitations in terms of novelty and direct clinical application.

Your feedback has illuminated the challenges we face in studying lncRNAs, particularly given their low expression and complex functional interactions with genomes, RNAs or proteins. We fully acknowledge that the functional complexity and diversity of lncRNAs make their analysis challenging, and it is precisely this difficulty that has led to their potential clinical applications being largely unexplored.

Our enrichment analysis was performed on genes associated with our m7G-associated lncRNAs. In a sense, it can be seen as an indirect way of performing a 'combined enrichment analysis' involving m7G modifications and lncRNA expression. Your observation that they provide stronger prognostic value in combination rather than individually has been enlightening, suggesting the existence of complex interactions in the cellular machinery.

Your valuable suggestion to further analyse the commonalities (both structural and functional) between m7G and lncRNAs has been taken into serious consideration. We agree that such an investigation would be valuable and a worthwhile direction for future research. However, given the current understanding of the molecular mechanisms of lncRNAs, and despite our comprehensive review of the literature and databases, we regretfully have not identified the necessary databases or other analytical methods to effectively perform such analyses.

We sincerely apologize if this aspect of our study does not meet your expectations and would like to assure you that your suggestion will serve as a critical direction for our future work.

We deeply appreciate your constructive feedback and wish to convey our genuine regret for any shortcomings in our work. We hope that our response has satisfactorily addressed your concerns, and we look forward to your continued guidance as we strive to improve our manuscript.

Response to Reviewer #2: 

Q1. How to identify a total of 47 m7G-related genes in GeneCards ?

Reply:Thank you for your comment regarding the identification of m7G-related genes.We employed the GeneCards database, a comprehensive, user-friendly compendium of annotated human genes, for this purpose. The search was conducted using the keyword "m7G" to capture any genes related to this modification.

GeneCards was selected due to its broad scope, which integrates gene-centric data from approximately 150 web sources, including genomic, transcriptomic, proteomic, genetic, clinical, and functional information.

Once the list was generated, we manually reviewed the results to ensure that each gene was indeed related to m7G modification based on its annotation or relevant literature. This process resulted in a list of 47 m7G-related genes.

Q2. Is there some references about "|R2| > 0.3 and P <0.001 were considered to indicate m7G-related lncRNAs" in materials and methods? Supply more details about indentification of M7G-related lncRNAs.

Reply:

Thank you for your comments on our criteria for determining m7G-related lncRNAs.

We appreciate your request for clarification regarding the thresholds we used: "|R2| > 0.3 and P <0.001". Our criteria of "|R2| > 0.3 and P <0.001" were informed by previous studies (PMID: 34488540, PMID: 32964050, PMID: 34869304, PMID: 35754819, PMID: 35846130, PMID: 35692827), which used similar thresholds to identify significant correlations. We have now cited these references in our revised manuscript.

At the same time we have added more details based on your suggestions, and in the revised manuscript it reads: "A systematic bioinformatics approach was employed to identify m7G-related long non-coding RNAs (lncRNAs). Expression profiles for m7G-modified genes and lncRNAs were obtained from our dataset. We then computed the Pearson correlation coefficient between expressions of m7G-modified genes and lncRNAs. An absolute Pearson correlation coefficient (|R^2|) greater than 0.3 and a P-value less than 0.001 were deemed significant. This strict criterion ensured the identification of lncRNAs exhibiting the strongest associations with m7G-modified genes. The method captured both positive and negative correlations. Finally, significantly correlated m7G-related lncRNAs were further investigated for potential biological functions and clinical relevance. Cytoscape 3.8.2 software was next used to visualize coexpression networks. "

Q3. Supply more details about "Development and validation of the nomogram" in materials and methods

Reply:We appreciate your insightful feedback and will include these details in our revised manuscript to provide a more comprehensive understanding of our methodology in the development and validation of the nomogram.

we have added more details based on your suggestions, and in the revised manuscript it reads: "Initially, a multivariate Cox regression analysis was performed, incorporating the risk score and clinicopathological parameters, to identify independent prognostic factors. These factors were then used to construct the prognostic nomogram using the "rms" package in R. For the development of the nomogram, each individual prognostic factor identified through the Cox regression analysis is assigned a weighted score in the nomogram, proportional to the magnitude of its regression coefficient. The sum of these scores forms the Total Points, which can be converted to a probability of survival at specified time points. Furthermore, calibration plots were generated to visually assess the predictive accuracy of the nomogram. In these plots, the x-axis represents the predicted survival probability by the nomogram, and the y-axis represents the actual observed survival. A 45-degree line in the plot represents perfect calibration, where predicted probabilities match the actual outcomes. The closer the calibration curve is to this line, the better the predictive accuracy of the nomogram."

Q4. Is an FDR of less than 0.25 considered significance in materials and methods ? Supply more details about "Gene set enrichment analysis"

Reply:Thank you for your comment. In our study, we elected to use a threshold of an FDR q-value less than 0.25 and a P-value less than 0.05 to determine significant enrichment. This decision was informed by our intention to incorporate a comprehensive range of potential biologically meaningful pathways or functions during the initial screening stage. We aimed to balance stringency with inclusivity in our analysis to ensure that we did not prematurely exclude any potentially significant biological insights.

we have added more details based on your suggestions, and in the revised manuscript it reads: "To investigate the biological pathways of the subgroups, we further performed gene set enrichment analysis (GSEA) for functional enrichment analysis. We employed the GSEA software from the Broad Institute, which allows the use of publicly available gene set databases, such as the Molecular Signatures Database (MSigDB). When performing GSEA, we first ranked our gene expression data, with this ranking based on the differential expression of genes across the different subgroups. Subsequently, we calculated an enrichment score (ES) for each gene set, reflecting the distribution of that set's genes within the ranked list. To determine the significance of the enrichment for each gene set, we generated a null distribution by random permutations of the original data and calculated a normalized enrichment score (NES) and normalized P-value for each set against this distribution. We then corrected these P-values to q-values using the False Discovery Rate (FDR) method to control for the probability of false discoveries when conducting multiple hypothesis tests."

Q5. Supply more details about "Prediction of response to immunotherapy" in materials and methods

Reply:I appreciate your request for further clarification regarding our methodology for predicting the response to immunotherapy. we have added more details based on your suggestions, and in the revised manuscript it reads: " The Tumor immune dysfunction and exclusion TIDE method is designed to model two primary mechanisms of tumor immune evasion that can affect the response to immunotherapy: the dysfunction of tumor-infiltrating cytotoxic T lymphocytes (CTLs) and the prevention of CTL infiltration. This model uses pre-existing transcriptomic data to generate predictions about the likelihood of response to immune checkpoint blockade treatment32. In our study, we obtained transcriptomic data of HCC patients from TCGA and stratified these patients into different risk groups based on their expression profiles. We then applied the TIDE method to each of these risk groups. The TIDE method uses the gene expression profile to calculate a TIDE score, which reflects the likelihood that the cancer will respond to immunotherapy. The TIDE prediction model is based on the principle that if the expression of a particular set of genes (known as a gene signature) associated with T cell dysfunction and exclusion is high, the tumor is more likely to evade the immune system, and hence the patient is less likely to respond to immune checkpoint blockade therapy."

Q6. Some minor mistakes need to be noted, such as "CO2" , "2 -△△Ct" , "As illustrated in Fig.B-D"

Reply: Thank you for your careful review and attention to detail. I appreciate your pointing out these minor errors. We have amended "CO2" to "CO2", "2 -△△Ct" to "2^-ΔΔCt" and "as illustrated in Fig.B-D" to "As illustrated in Fig. 5B-D".

---

## [Editor Report · Decision Letter 1]

21 Jul 2023

Identification and validation of a m7G-related lncRNA signature for predicting the prognosis and therapy response in hepatocellular carcinoma

PONE-D-23-06969R1

Dear Dr.Shan Li

We’re pleased to inform you that your manuscript has been judged scientifically suitable for publication and will be formally accepted for publication once it meets all outstanding technical requirements.

Kind regards,

Afsheen Raza, PhD

Academic Editor

PLOS ONE
---

## [Editor Report · Acceptance letter]

26 Jul 2023

PONE-D-23-06969R1 

Identification and validation of a m7G-related lncRNA signature for predicting the prognosis and therapy response in hepatocellular carcinoma 

Dear Dr. Li:

I'm pleased to inform you that your manuscript has been deemed suitable for publication in PLOS ONE. Congratulations! Your manuscript is now with our production department. 

Kind regards, 

on behalf of

Dr. Afsheen Raza 

Academic Editor

PLOS ONE